# Carbon/Basalt Fibers Hybrid Composites: Hybrid Design and the Application in Automobile Engine Hood

**DOI:** 10.3390/polym14183917

**Published:** 2022-09-19

**Authors:** Yongfeng Pu, Baichuan Liu, Guilian Xue, Hongyu Liang, Fangwu Ma, Meng Yang, Guangdong Tian

**Affiliations:** 1State Key Laboratory of Automotive Simulation and Control, Jilin University, Changchun 130022, China; 2Automobile Research Institute of China Heavy Duty Automobile Group Co., Ltd., Jinan 250031, China; 3School of Mechanical-Electrical and Vehicle Engineering, Beijing University of Civil Engineering and Architecture, Beijing 100044, China

**Keywords:** carbon/basalt hybrid composites, low-velocity impact properties, hybrid design, carbon/basalt hybrid composites hood

## Abstract

The low-velocity impact properties and the optimal hybrid ratio range for improving the property of hybrid composites are studied, and the application of hybrid composites in automobile engine hoods is discussed in this paper. The low-velocity impact properties of the hybrid composite material are simulated under different stacking sequences and hybrid ratios by finite element simulation, and the accuracy of the finite element model (FEM) is verified through experiments. Increasing the proportion of carbon fiber (CF) in the hybrid layer and placing the basalt fiber (BF) on the compression side can improve the energy absorption capacity under low-velocity impact loads. CF/BF hybrid composite hoods are optimized based on the steel hood and the low-velocity impact performance of the hybrid composite. The BCCC layer absorbs the most energy under low-velocity impact loads. Compared with CFRP, the energy absorbed under 10 J and 20 J impact energy is increased by 26.1% and 14.2%, respectively. Through the low-velocity impact properties of hybrid composites, we found that placing BF on the side of the load and keep the ratio below 50%, while increasing the proportion of CF in the hybrid laminate can significantly improve the property of the hybrid laminate. The results show that the stiffness and modal properties of the hybrid composite can meet the design index requirements, and the pedestrian protection capability of the hood will also increase with the increase in the proportion of BF.

## 1. Introduction

Carbon fiber reinforced resin matrix (CFRP) composites have significant advantages of being lightweight, having high heat resistance, good corrosion resistance, and high strength [1,2,3]. Significant attention has been gained in CFRP in recent decades, and now has been widely used in lightweight automobile development, the aerospace industry, and other high-performance industries; these all due to their excellent mechanical properties [4,5,6,7]. However, the problems of high brittleness and the high cost of CF have always restricted its popularization in automobile applications [8,9,10]. Compared with single-fiber composite materials, hybrid composite materials can be more well-designed, and also can make up for the defects of single-fiber materials and broaden the application range of composite materials [11,12,13].

Several research papers on the performance of hybrid composite materials have been conducted at present, which mainly focused on the hybrid performance of different fibers. Research on the low-velocity impact properties of hybrid composites are particularly important as the force form of composite materials is more complicated. In recent years, CF and glass fiber reinforced hybrid composites have been studied as shown in Table 1.

The results showed that placing CF on the impact surface would reduce the propagation of damage, while glass fiber [20,21] placed on the impact surface would cause greater damage. The natural fiber [22] such as BF has a higher breaking elongation, not lower than the tensile and compressive mechanical properties of glass fiber, good corrosion resistance and chemical resistance, as well as low price [23,24,25]; this was investigated to explore the mechanical properties of the hybrid composites by carbon/basalt [26,27]; moreover, it is a new type of environmentally friendly material.

In this paper, CF and BF are combined into a hybrid material, an epoxy resin is used as the matrix material [28,29,30]. Through studying the hybrid ratio range and layup method to improve the performance of the hybrid material, the application of the hybrid composite material on the automobile engine hood is discussed and analyzed. The low-velocity impact properties and the optimal hybrid ratio range for improving the property of hybrid composites are studied. This research is of great significance to automotive fields.

## 2. Methods

### 2.1. Materials and Processing

The reinforcement materials CF are purchased from TAIRYFIL Carbon Fiber (Taipei, Taiwan, China) (TC-35-12K) and BF are obtained from Jilin Tongxin Basalt Technology Co., Ltd. (Tonghua, China) (264tex), and the matrix material epoxy resin are from Kunshan Truetime Epoxy Complex Co., Ltd. (Kunshan, China) (5113-81A/5113-94B). The specific material parameters are shown in Table 2.

Ten different hybrid composite laminates are designed by different hybrid ratios and stacking sequences to investigate the optimal hybrid ratio range. The specific stacking sequences are shown in Figure 1. Each laminate is composed of 16 layers of material, and every 4 layers from the surface layer are composed of the same material. It means that the whole ply can be divided into 4 units. The first unit and the last unit from the surface layer are at the same angle (0°) and the two units in the middle are at the same angle (90°).

The recommended thickness and number of impact laminates were set according to ASTM D7136. The low-velocity impact tests were designed into ten different patterns of mixing ratio and stacking sequence, and the laying angle is 0° and 90°. Meanwhile, BF with good toughness is placed on the surface layer.

### 2.2. Low-Velocity Impact Test

Low-velocity impact tests were carried out at room temperature on standard ASTM D6641, at a speed of 2 mm/min with the impact energy are 10 J and 20 J. Specimens were molded into a dumbbell shape to 140 mm × 12 mm × 4 mm, respectively, for the tests. Each specimen was tested a minimum of five times per sample. Figure 2 is the test device of low-velocity impact tests.

### 2.3. Damage Degradation Model of Composite Materials

The damage degradation model of composite materials includes strength criterion and degradation criterion. In order to improve the accuracy of finite element simulation, this paper uses cohesive elements to simulate the performance of fiber layers.

#### 2.3.1. Strength Criterion

The theoretical damage degradation model of composite materials was established and verified while the performance prediction and damage mechanism of hybrid composites were explored. By combining the strain-based Chang-Chang fiber strength theory [31,32], the fracture toughness stiffness degradation theory, and the cohesive element layering theory, a composite damage theory model was established for theoretical analysis.

The strain-based Chang-Chang strength criterion is shown in the formula.

(1)Fiber tensile failure


(1)
$${\left(\frac{\varepsilon_{11}}{X_{T}^{\varepsilon }}\right)}^{2}+{\left(\frac{\varepsilon_{12}}{S_{12}^{\varepsilon }}\right)}^{2}+{\left(\frac{\varepsilon_{13}}{S_{13}^{\varepsilon }}\right)}^{2}\geq 1$$


(2)Fiber compressive failure


(2)
$${\left(\frac{\varepsilon_{11}}{X_{C}^{\varepsilon }}\right)}^{2}+{\left(\frac{\varepsilon_{12}}{S_{12}^{s}}\right)}^{2}+{\left(\frac{\varepsilon_{13}}{S_{13}^{\varepsilon }}\right)}^{2}\geq 1$$


(3)Matrix tensile failure


(3)
$${\left(\frac{\varepsilon_{22}}{Y_{T}^{\varepsilon }}\right)}^{2}+{\left(\frac{\varepsilon_{12}}{S_{12}^{\varepsilon }}\right)}^{2}+{\left(\frac{\varepsilon_{23}}{S_{23}^{\varepsilon }}\right)}^{2}\geq 1$$


(4)Matrix compressive failure


(4)
$${\left(\frac{-\varepsilon_{22}E_{22}}{2S_{12}^{\varepsilon }G_{12}}\right)}^{2}+\frac{\varepsilon_{22}}{Y_{C}^{\varepsilon }}\left[{\left(\frac{E_{22}Y_{c}^{\varepsilon }}{2G_{12}S_{12}^{\varepsilon }}\right)}^{2}-1\right]+{\left(\frac{\varepsilon_{12}}{S_{12}^{\varepsilon }}\right)}^{2}\geq 1$$


In the formula, *ε* is the internal strain in the corresponding direction during the stress process of the material, $X_{T}^{\varepsilon }$ and $X_{C}^{\varepsilon }$ are the ultimate strains in the tensile and compression directions of the fiber, $Y_{t}^{\varepsilon }$ and $Y_{C}^{\varepsilon }$ are the ultimate strains in the tensile and compression directions of the matrix, $S_{12}^{\varepsilon }$, $S_{13}^{\varepsilon }$, $S_{23}^{\varepsilon }$ is the ultimate shear strain in and out of the corresponding plane.

#### 2.3.2. Degradation Criterion

The stiffness degradation model in this paper adopts a linear degradation mode, which directly degrades the material stiffness to 0 or a minimum value compared to the parameter degradation, which is closer to the real degradation mode of the material. At the same time, the element characteristic length is introduced to avoid the influence of the element size on the failure process.

Assuming that $\varepsilon_{ii}$ is the strain of the fiber or matrix at a certain moment, the damage variable can be expressed as,
(5)$$d_{ii}\left(\varepsilon_{ii}\right)=\frac{\varepsilon_{ii}^{F}}{\varepsilon_{ii}^{F}-\varepsilon_{ii}^{0}}\left(1-\frac{\varepsilon_{ii}^{0}}{\varepsilon_{ii}}\right)$$

$\varepsilon_{ii}^{0}$ is the initial strain at which the fiber or matrix is damaged, and $\varepsilon_{ii}^{F}$ is the final strain at which the fiber or matrix is completely destroyed. These two values can be calculated by Equation (6).
(6)$$\begin{array}{c} \varepsilon_{11}^{Ft}=\frac{2G_{1c}^{t}}{X^{T}l^{*}},\varepsilon_{11}^{Fc}=\frac{2G_{1c}^{c}}{X^{C}l^{*}}\\ \varepsilon_{22}^{Ft}=\frac{2G_{2c}^{t}}{Y^{T}l^{*}},\varepsilon_{22}^{Fc}=\frac{2G_{2c}^{c}}{Y^{C}l^{*}} \end{array}\}$$

In the formula, $G_{1c}^{t}$ is the tensile fracture toughness in the fiber direction of the material, $G_{1c}^{c}$ is the compressive fracture toughness in the fiber direction of the material, $G_{2c}^{t}$ is the tensile fracture toughness in the matrix direction of the material, $G_{2c}^{c}$ is the compressive fracture toughness in the matrix direction (the thickness direction, where the matrix is perpendicular to the fiber) of the material, and $l^{*}$ is the characteristic length of the element, which is defined in ABAQUS (ABAQUS 6.14), and has different values according to the size of the element.

#### 2.3.3. Degradation Criterion

The interlayer zone is simulated by cohesive element, and the secondary stress criterion is used to judge the initiation of the interlayer damage. The normal stress and two tangential shear stresses of the cohesive element are represented by $t_{n}$, $t_{t}$, $t_{s}$ while $t_{n}^{\max}$, $t_{t}^{\max}$, $t_{s}^{\max}$ are the ultimate strength in the corresponding direction, as shown in the formula. (7)$${\left\{\frac{\langle t_{n}\rangle }{t_{n}^{\max}}\right\}}^{2}+{\left\{\frac{t_{s}}{t_{s}^{\max}}\right\}}^{2}+{\left\{\frac{t_{t}}{t_{t}^{\max}}\right\}}^{2}=1$$

In the formula, (8)$$\langle t_{n}\rangle =\left\{\begin{array}{c} t_{n},t_{n}>0\\ 0,t_{n}<0 \end{array}\right\}$$

According to the energy release rate of the element, the damage expansion criterion of the interlayer element in the mixed mode proposed by Benzeggagh and Kenane [33] is used to simulate the delamination expansion of the laminate, as shown in Equation (9), when *G_c_* > 1, the material has undergone delamination failure. (9)$$\{\begin{array}{c} G_{c}=G_{n}^{c}+\left(G_{s}^{c}-G_{n}^{c}\right){\left(\frac{G_{S}}{G_{T}}\right)}^{\eta }\\ G_{S}=G_{s}+G_{t},G_{T}=G_{n}+G_{S} \end{array}$$

*G_n_*, *G_s_*, *G_t_* are the energy release rates of type I opening, type II slip, and type III tearing, respectively, while $G_{n}^{c}$, $G_{s}^{c}$, $G_{t}^{c}$ are the corresponding critical strain energy release rates. For the epoxy-based composite material, it is generally 1~2, and the value is 1.5 in this article.

### 2.4. Finite Element Model (FEM)

Explicit dynamic finite element analysis was used to simulate the transient response of composites under impact by ABAQUS. The dropping hammer impact test model was established according to the conditions of the falling hammer impact test. The gravity action of the device and the fixture device, which mainly composed of punch and laminated plate, were ignored to simplify the model. Each parameter corresponds to the test conditions. The steel engine hood was modeled before performing the mid-surface operation to reduce the calculation time of the model.

The parameters correspond to the test conditions, and the specific modeling process is as follows:(1)Model building: first build the laminate and punch model. The size of the laminate is 125 mm × 100 mm, the thickness of the single layer is 0.2 mm, the thickness of the cohesive force unit is 0.02 mm, and the total thickness of the laminate is 3.5 mm;(2)Material properties: the basic parameters of the material are obtained from the material mechanical properties test. The fracture toughness of carbon fiber and basalt fiber is provided by the manufacturer. The ply and stacking method of the laminate is the same as that of the experimental design;(3)Assembly: to reduce the analysis time, the punch and the laminate are in contact with the laminate at a distance of 0 mm;(4)Contact: the punch is set as a rigid body due to its high stiffness and the main concern is the response of the laminate under low-velocity impact;(5)Constraints: to simplify the model, constrain the degrees of freedom of the elements around the laminate in 6 directions, and apply a vertical downward velocity load to the punch. Under the impact energy of 10 J and 20 J, the speed of the punch in contact with the laminate is 1898 mm/s and 2697 mm/s, respectively;(6)Mesh: the mesh of the laminate is offset and refined, so that the mesh in the impact area is finer to improve the simulation accuracy;(7)Calculation: use the composite damage degradation model established for calculation.

### 2.5. Optimal Design

#### 2.5.1. Hybrid Engine Hood Optimization

The steel engine hood was modeled before performing the mid-surface operation to reduce the calculation time of the model. An element size of 10 mm was used while the proportion of triangular elements should be below 5%. There are 37,511 quadrilateral elements and 1807 triangular elements, accounting for 4.39% of all elements after the model was meshed, which meets the mesh quality requirements.

#### 2.5.2. CF Hood Optimization

Due to the different manufacturing processes of composite materials, it is necessary to improve its details on the basis of the steel engine. Cancel the electrophoresis hole and soundproof cotton fixing hole of the original model, and use structural adhesive to connect the inner and outer panels.

#### 2.5.3. Ply Cut Shape Optimization

According to the influence of various working conditions of the engine hood, the parameters for optimizing the ply cut shape are as follows: the variable is the thickness of each orientation layer; the goal is to get the smallest weighted flexibility (largest stiffness) under each working condition, and the constraints are to keep the volume fraction less than 0.3 [34,35,36], the proportion of each layering direction between 10% and 60%, and the symmetrical distribution of ±45° and the same thickness.

The mathematical expression for the optimization of the ply cut shape is (10)$$\begin{array}{c} \min C_{W}=\overset{i=7}{\underset{i=1}{\sum }}W_{i}C_{i}\\ s.t.\{\begin{array}{l} Vol\leq 0.3\\ 0.1\leq P_{\theta }\leq 0.6 \end{array} \end{array}\}$$

In the formula, *C_W_* is the weighted compliance; *W_i_* is the weighted coefficient; *C_i_* is the compliance; *i* is the number of weighted operating conditions, which are the seven different operating conditions; *Vol* is the volume fraction of the inner and outer plates; *P* is the layup ratio, which refers to the proportion of plies at different angles to the total layup; *θ* is the orientation, and the value range is 0°, 45°, −45°, 90°.

#### 2.5.4. Layer Thickness Optimization

After the ply cut shape optimization, 16 initial layups with different orientations are optimized into 64 small layers, and each small layer needs to be adjusted through layup thickness optimization, so that the small layers that contribute to the stiffness are retained. The variable of the optimization is thickness distribution of inner and outer panels. The target is to get the smallest mass of the engine hood. The rigidity performance of the engine hood is not lower than that of steel, and the thickness of the ply can be 0.2 mm, and the number of plies of ±45° are equal; the proportion of each ply direction is 10~60%, which ensures the symmetrical distribution of ±45° with the same thickness.

The mathematical expression for layup thickness optimization is (11)$$\begin{array}{l} \min mass\\ s.t.\{\begin{array}{l} u_{i}-u_{z0}\leq 0\\ f_{0}-f\leq 0\\ 0.1\leq P_{\theta }\leq 0.6 \end{array} \end{array}$$

In the formula, *mass* is the mass of the engine hood; *u_z0_* is the displacement of the clamp action point of the steel engine hood; *i* is the number of weighted working conditions, which are the seven different working conditions; *f*_0_ is the first-order mode of steel engine hood; *u_i_* and *f* are the displacements and first-order mode of the CF composite material engine hood.

#### 2.5.5. Stacking Sequences Optimization

Composite materials have a larger design space than metal materials on account of their unique layup orientation and stacking sequences. The optimization of the layup sequence is carried out based on the results of thickness optimization and only by optimizing the ply orientation and sequence. In this step, add two design constraints to the optimization of the ply sequence according to the requirements of the process. The number of consecutive plies at a single orientation does not exceed two and the surface of the outer panel is laminated with ±45°.

The composite material optimization adopts the segmented optimization method. The cutting shape of each layer of the laminate is determined firstly, and the optimization results are optimized for the layer thickness. Finally, the orientation and stacking sequence are optimized according to the process and manufacturing constraints. The specific steps are shown in Figure 3.

#### 2.5.6. Pedestrian Head Impact Test

The pedestrian head impact test of the engine hood is conducted based on the C-NCAP2018. The adult head type and the child head type hit the corresponding area of the engine hood at an impact angle of 65° and 50°, respectively, and the acceleration change in the impactor is read through the three-axis accelerometer installed in the middle of the head type impactor, thereby evaluating the safety performance through analyzing the acceleration. The evaluation of the pedestrian protection function of the front engine room cover is based on the HIC value [37], as shown in the Formula (12). (12)$$\mathrm{HIC}=\max\left({\left[\frac{1}{t_{2}-t_{1}}\int_{t_{1}}^{t_{2}}adt\right]}^{2.5}\left(t_{2}-t_{1}\right)\right)$$

## 3. Results and Discussion

Figure 4 shows the stress variation trend of CF/BF hybrid composite laminates with the impact process under the impact energy of 10 J.

The stress (Mises stress) at the bottom of the laminated plate is the largest, while the laminated material at the intermediate layer and the contact part is relatively small. The punch reaches maximum penetration depth and the contact force reaches the peak at t = 1.96 ms, then the punch bounces back until they are separated by the elastic action of the laminate. There is no contact relationship between the punch and the laminate at t = 6.58 ms, but there is residual stress in the laminate. The impact process lasts longer due to the small elastic modulus and strong toughness of BF.

Setting CF on the impact surface, under different hybrid ratios and stacking sequences, can significantly increase the peak impact force of the hybrid laminate, such as CBCC and CBBC, seen in Figure 5a. Similarly, laying CF on the side close to the impact surface can also increase the peak impact force of the hybrid layer. This can also be found from the comparison of BBCB, BBBC, BCCB, BCBC and BBCC.

Figure 5b shows the peak force trend curve and comparison diagram of the FEM and the test. It also shows that the peak force is closely related to the first layer of fiber types. Compared with pure basalt fiber composite materials, the peak force of the first layer of carbon fiber laminates such as CBBC and CBCC has been greatly improved by 20.1% and 14.9%, respectively. However, the change in the peak force amplitude is small, even slightly lower when the first floor is BF. For example, BBBC at 10 J and 20 J impact energies are reduced by 6.1% and 7.2%, respectively, compared with pure BF laminates. The CF with the highest stiffness also has the highest impact resistance, with a peak force of 21.7 percent higher than that of pure BF. Under different mixing ratios, the errors of this simulation data and experimental data are similar, and the error rates of peak force value are all within 5%.

When pedestrians collide with the engine hood of the car, the cushioning effect and energy absorption capacity of the structure on the pedestrian’s head are more important. Figure 6a shows the energy absorption capacity of different stacking sequences and the energy absorbed by the laminate obtained from the FEM and test is shown in Figure 6b. The variation trend of different hybrid ratios is basically the same, and the error of the peak force value is all within 5%. Setting BF on the impact side can increase the energy absorption capacity of the laminate. Since the vehicle engine hood needs to cushion the pedestrian’s head during the collision with the pedestrian’s head, it is necessary to choose a layer that absorbs more energy, and the BCCC absorbs the most energy at both 10 J and 20 J impact energy. However, under the impact energy test of 10 J, the BBBB and BBCB layers absorbed less energy, which may be caused by the interlayer defects. The result can be concluded that keeping the hybrid ratio of CF below 50%, and setting BF on the impacted side can improve the overall properties of hybrid composites.

The composite materials have better specific strength performance, and it is first necessary to ensure that the composite material engine hood meets the rigidity conditions and then check the strength property when using the equivalent design method to optimize the design of composite materials. The young modulus, Poisson ratio and density of structural adhesive are 1515 MPa, 0.41 and 1.4 × 10^−9^ t mm^−3^, respectively. The models of inner and outer panels are shown in Figure 7.

The thickness distribution of the inner and outer panel can be obtained after optimization, seen in Figure 8. According to the influence of various working conditions of the engine hood, the specific process for optimizing the ply cut shape is as follows:(1)Optimization variable: the thickness of each orientation layer.(2)Optimization goal: the smallest weighted flexibility (largest stiffness) under each working condition.(3)Optimization constraints: keep the volume fraction less than 0.3, the proportion of each layering direction between 10% and 60%, and the symmetrical distribution of ±45° and the same thickness.

The most areas of the outer plate are thinner, and only the middle and edge positions have thicker areas (the red areas). The maximum thickness is 4.673 mm. The red area of the inner panel is larger than that of the outer panel, with a maximum thickness of 4.8 mm. After optimization, the mass of the engine hood was reduced from 25.02 kg in the initial design to 14.75 kg. The thickness of the cut-out shape needs to be optimized to make the inner and outer panel layer more uniform when the optimized engine hood of the lay-up cut shape does not meet the manufacturing requirements.

The thickness changes of the inner and outer panels in each iteration step during the optimization of the layer thickness are shown in Figure 9 and Figure 10. The specific process of layer thickness optimization is as follows:(1)Optimization variable: thickness distribution of inner and outer plates;(2)Optimization objective: the mass of the front cabin cover is the smallest;(3)Optimization constraints: the rigidity performance of the front cabin cover shall not be lower than that of steel; the number of layers with a thickness of 0.2 mm and ±45° can be manufactured; the proportion of each layer direction is 10~60%, and ensure ±45° symmetrical distribution and the same thickness.

The figures show that the thickness of the outer plate after optimization is 1.6 mm, while the inner panel is 1.4 mm, and the thickness of the inner plate after optimization is 1.4 mm, with eight and seven layers, respectively. The lay orientations are [0°/45°/90°/−45°/−45°/90°/45°/0°] and [0°/45°/−45°/−45°/90°/45°/0°]. The mass of the model is reduced from 14.75 kg to 9.445 kg, after layer thickness optimization.

The composite materials, which have unique layup orientation and stacking sequences, have a larger design space than metal materials. The optimization of the layup sequence is carried out based on the results of thickness optimization and only optimizing the ply orientation and sequence. Two design constraints were added to optimize the ply sequence according to the requirements of the process:(1)The number of consecutive plies at a single orientation does not exceed two;(2)The surface of the outer panel is laminated with ±45°.(3)Each column in the figure represents the layup results of different iterative steps; 741 and 742 represent the outer and inner panels, respectively. The mass of the engine hood did not decrease for only the optimized sequence. The final optimized outer and inner layups are [−45°/45°/0°/90°/90°/0°/45°/−45°] and [0°/90°/45°/−45°/0°/45°/−45°].

Since the inner and outer panels of the engine hood are two different parts, replace the CF of the outer panel with BF at a layup ratio of 50%. The hybrid ratios are 12.5%, 25.0%, 37.5%, which are expressed by B1, B2, and B3, respectively. Table 3 shows the comparison of the analysis results of steel, CF and three hybrid fiber engine covers. The stiffness of the optimized CF structure is greater than that of the steel structure, which meets the design requirements.

In the Table 3, Condition 1 is the installation deformation condition; Condition 2 is the edge deformation condition; Condition 3 is the pull-down deformation condition; Condition 4 is the torsion deformation condition; Condition 5 is the impact deformation condition; Condition 6 is the modal analysis.

Despite it being lower than the steel structure under certain conditions (Condition 4 and B2, B3 under Condition 2.2), the stiffness of the hybrid structure also meets the requirements of the design index. The weight of CF is 42.1% lower than that of the steel structure, while the other three hybrid structures are also reduced by 40.3%, 38.8% and 37.1%, respectively. The application of composite materials for the design of engine hood has a very obvious lightweight advantage.

The working conditions of the tests are installation deformation analysis, edge deformation analysis, pull-down deformation analysis, torsional stiffness analysis and modal analysis. Figure 11a shows the manufacturing process of the engine hood, which is mainly divided into making the mold, laying fibers, injecting resin, curing and spotting and polish.

The corresponding test pictures and test results are shown in Figure 11b and Table 4. Under four different loading conditions, the maximum displacement obtained by the test is higher than the simulation result, and the error is within 10%, and the error value = (test value − simulation value)/test value × 100%. The main reason for the error is the uncertainty of the manufacturing process.

The engine hood is divided into adult head-shaped impact areas A1D, A2C, A2D, A3C, A3D, and children’s head-shaped impact areas are C2A, C2B, C3A, C3B, C1D, C2C, C2D, C3C, C3D according to the standard, seen in Figure 12a. As the engine is a symmetrical structure, only the area on one side of the middle line is simulated for head impact. The HIC values of different laminates are shown in Figure 12b. Different impact positions have different HIC values, and the values of A1D, C1D, C2A and C3C are all above 1300. The middle part of the side edge of the engine hood is fully constrained during the pre-processing of the model to simulate the effect of fender, therefore the cushioning effect on the head shape is small which results in high HIC.

The position of the reinforcement plate and the latch in the model, such as the A1D and C3D areas, to ensure the rigidity of the engine hood and facilitate installation, will also cause the increase in HIC. The HIC in the middle of the engine hood (e.g., the A3D and C3B areas) is smaller; the head-shaped impactor only collides with the inner and outer plates, which has a better cushioning effect on the impactor. The HIC value is the lowest among all piles when three layers of basalt fiber are set on the surface of the outer panel. Due to the influence of constraints, A1D, C1D, C2C and C3D are higher than 1000 but below 1200.

The ultimate goal of pedestrian head collision protection is to reduce the HIC value to meet the evaluation requirements of the regulations, but the maximum intrusion during the impact must be controlled to reduce the HIC. Increasing the hybrid ratio of BF can significantly reduce the HIC value, and also increase the amount of intrusion. The impact of the head-shaped impactor on the engine hood is a process of converting kinetic energy into internal energy from the perspective of energy absorption. The energy absorbed by the engine hood at 14 impact points with the increased proportion of BF is shown in Figure 12c.

The higher the energy absorbed in the same collision position, the lower the HIC value and the more obvious protective effect on pedestrians. However, if the different shapes of the inner panel at different collision positions have different abilities to limit impact deformation, then the HIC value of the engine hood of the same material cannot be determined based on the energy absorption capacity. For example, A3D and C3B are located in the middle area of the engine hood with the smallest HIC value, which is lower than the A1D and C1D points.

The energy absorption capacity of the engine hood basically increases with the increased proportion of BF, especially at the edge of the engine hood (the A1D and C1D), for the edge constraint of the laminate and the effect of the reinforcing plate, which causes greater stress at the collision position and leads to more energy absorption.

## 4. Conclusions

The low-velocity impact properties and the optimal hybrid ratio range for improving the property of hybrid composites are studied, and the application of hybrid composites in automobile engine hoods is discussed in this paper.

Research on low-velocity impact property has found that setting CF close to the impact surface could increase the peak impact force of the overall laminate, and setting BF on the impact surface can increase the energy absorption capacity. Among them, the BCCC layer absorbs the most energy under low-velocity impact loads. Compared with CFRP, the energy absorbed under 10 J and 20 J impact energy is increased by 26.1% and 14.2%, respectively. We found that the low-velocity impact properties of hybrid composites by placing BF on the side of the load and keeping the ratio below 50% while increasing the proportion of CF in the hybrid laminate can significantly improve the property of the hybrid laminate.

The optimized design and performance verification of the hybrid composite front nacelle cover was carried out. Taking the response values of the original steel front nacelle cover under different working conditions as the design boundary conditions, a model of the carbon fiber composite front nacelle cover was established and optimized. The accuracy of the model was verified by experiments. The CF/BF hybrid composite engine hood is optimized based on the steel structure. Results show that the engine hood hybrid with BF can meet the rigidity and modal requirements, and the quality of the engine hood is reduced by 40.3%, 38.8%, 37.1% compared with the steel structure.

The research on the pedestrian protection function of CFRP and CF/BF hybrid composite materials shows that with the increase in proportion of BF, the HIC value gradually decreases. The HIC value of the engine hood mixed with three layers of BF is the lowest, while the maximum intrusion is 22.4 mm, which will not collide with the parts in the engine cabin.

## Figures and Tables

**Figure 1 polymers-14-03917-f001:**
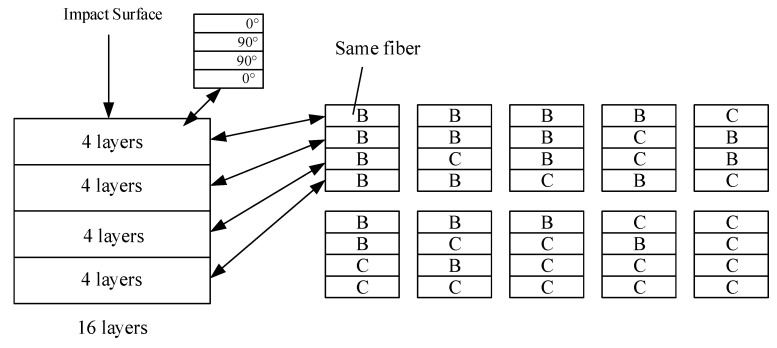
Schematic diagram of low-velocity impact laminate ply.

**Figure 2 polymers-14-03917-f002:**
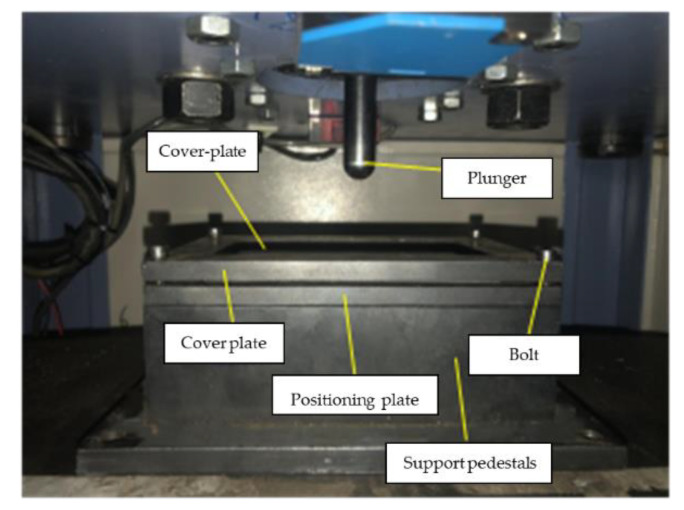
The test device of Low- velocity impact tests.

**Figure 3 polymers-14-03917-f003:**
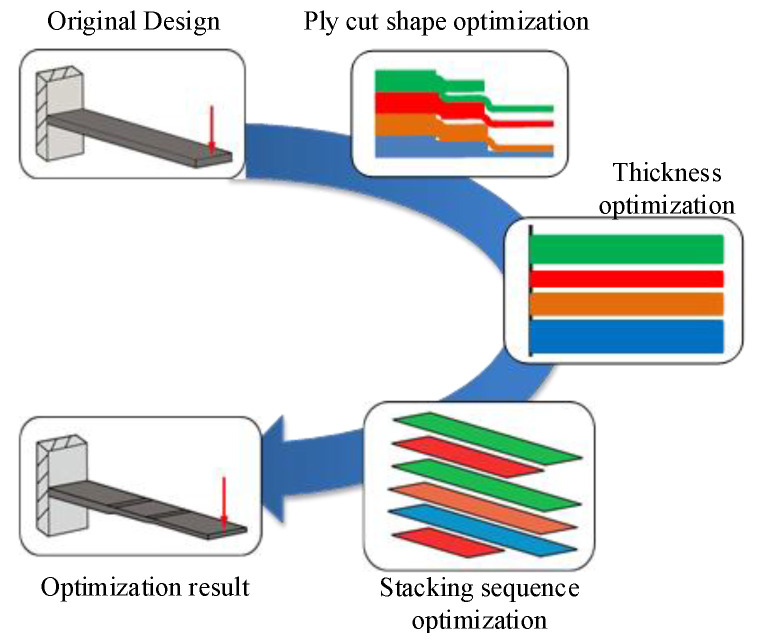
Steps of segmented optimization.

**Figure 4 polymers-14-03917-f004:**
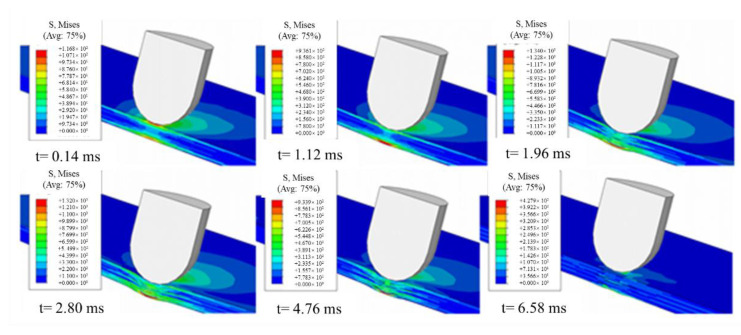
Impact history of CF/BF hybrid composite laminates.

**Figure 5 polymers-14-03917-f005:**
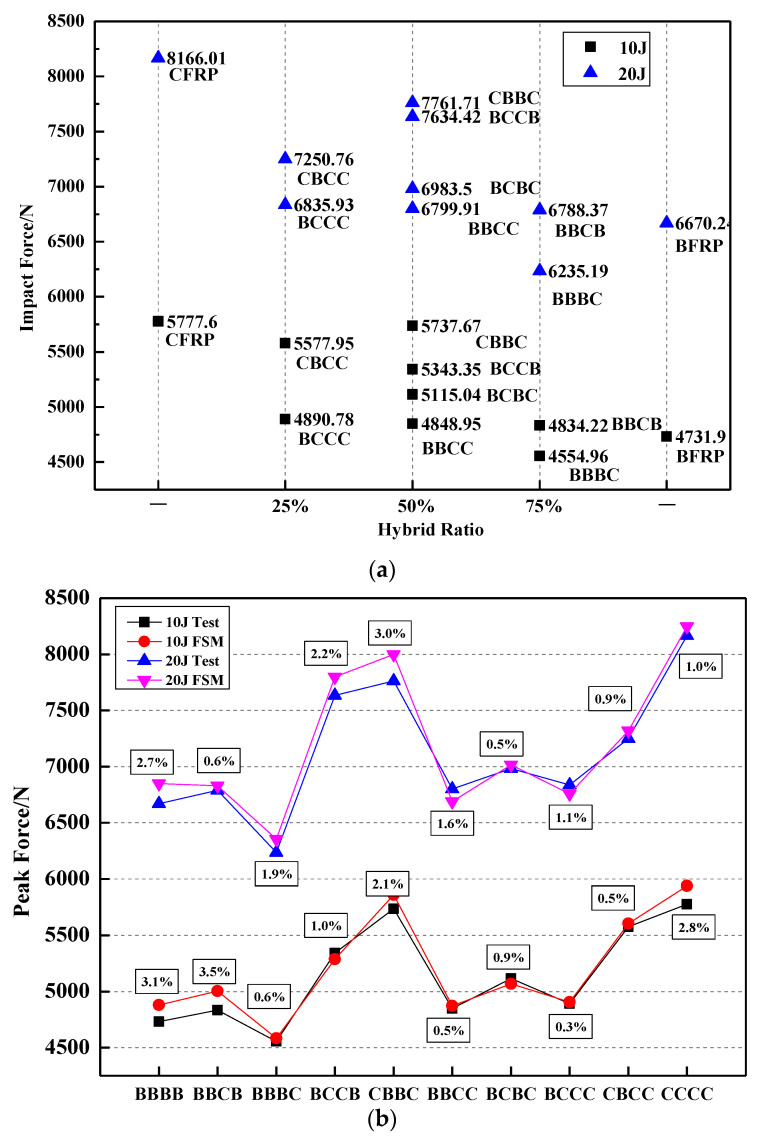
Peak force of (**a**) different hybrid ratio and (**b**) trend curve and comparison diagram of FEM and test.

**Figure 6 polymers-14-03917-f006:**
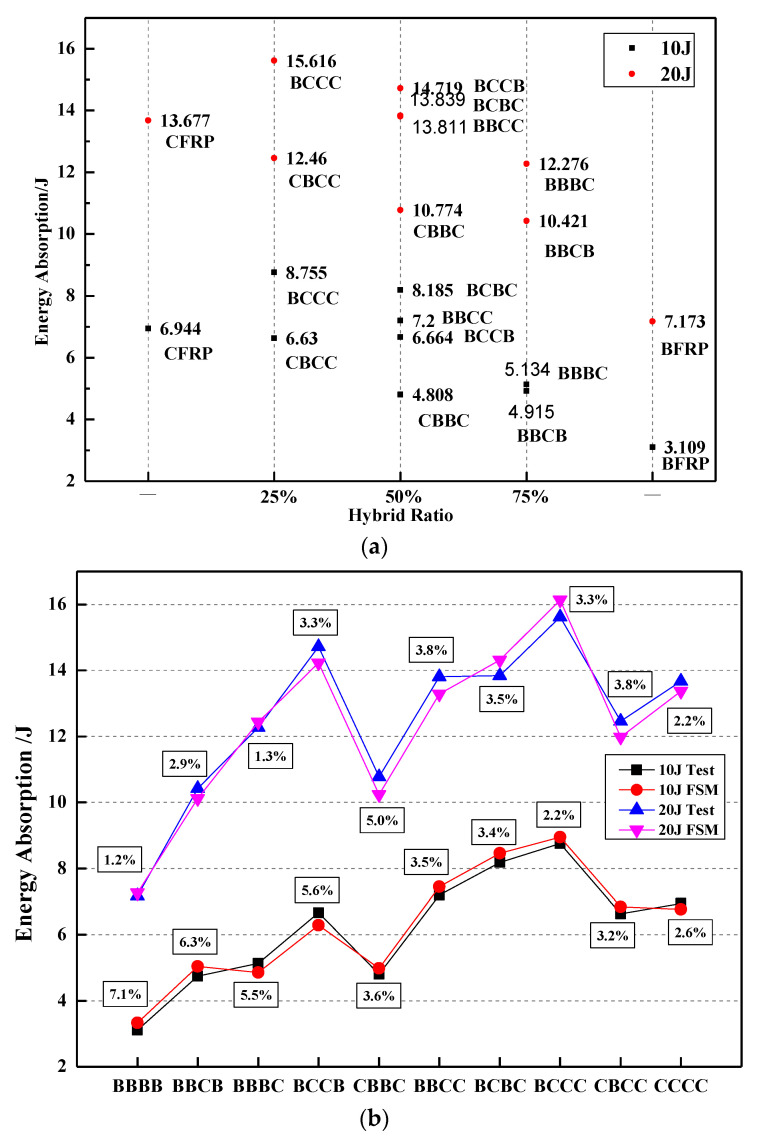
Energy absorption of (**a**) different impact laminates and (**b**) trend curve and comparison diagram of FEM and test.

**Figure 7 polymers-14-03917-f007:**
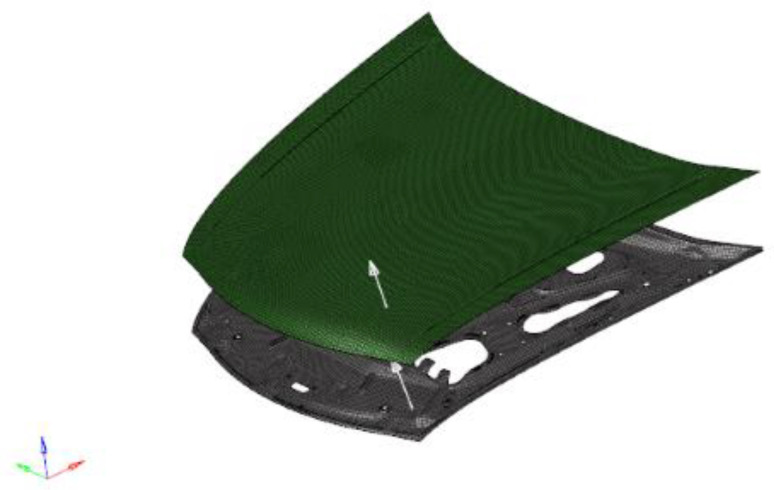
Model of CF/BF hybrid composites engine hood.

**Figure 8 polymers-14-03917-f008:**
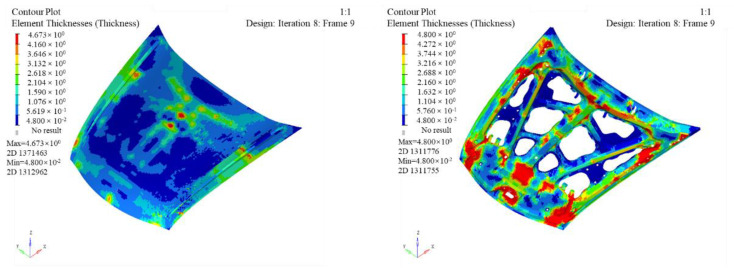
Cloud map of inner and outer panel thickness after optimization of ply cut shape.

**Figure 9 polymers-14-03917-f009:**
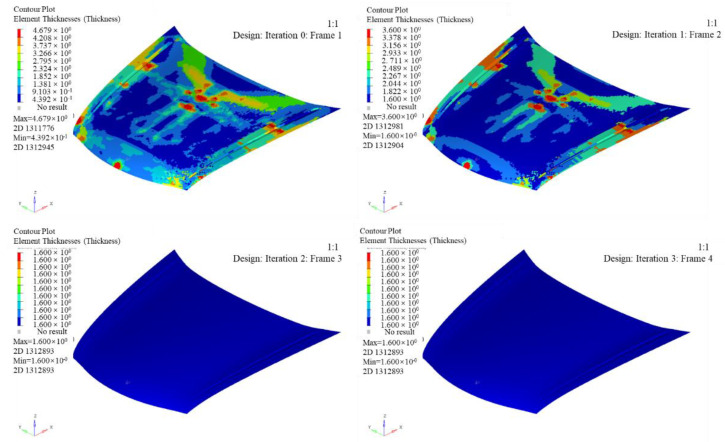
Contour map of outer plate thickness after thickness optimization.

**Figure 10 polymers-14-03917-f010:**
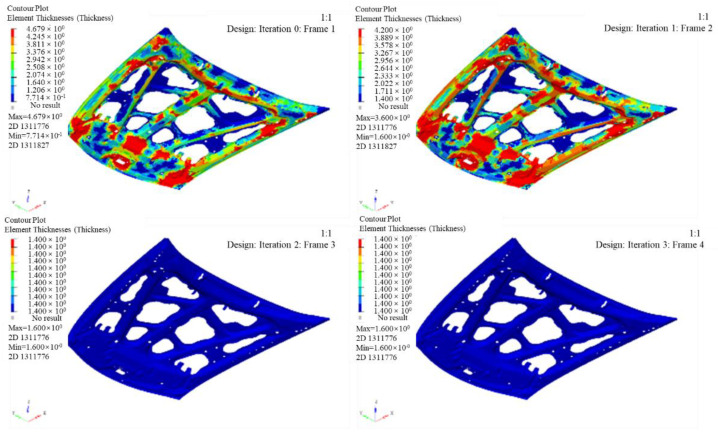
Contour map of inner plate thickness after thickness optimization.

**Figure 11 polymers-14-03917-f011:**
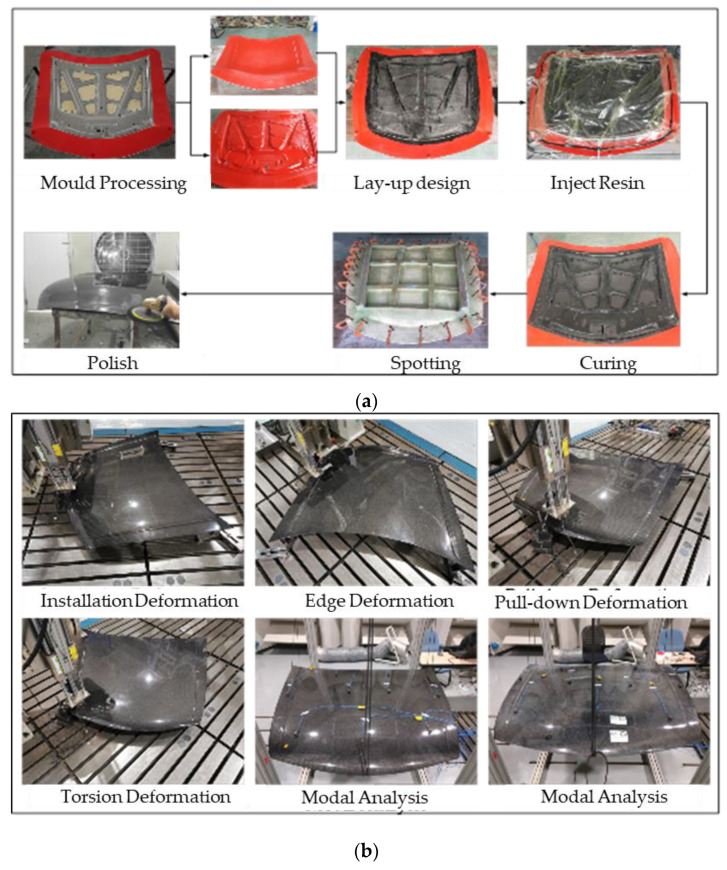
(**a**) engine hood manufacturing operation, (**b**) stiffness and modal tests of engine hood.

**Figure 12 polymers-14-03917-f012:**
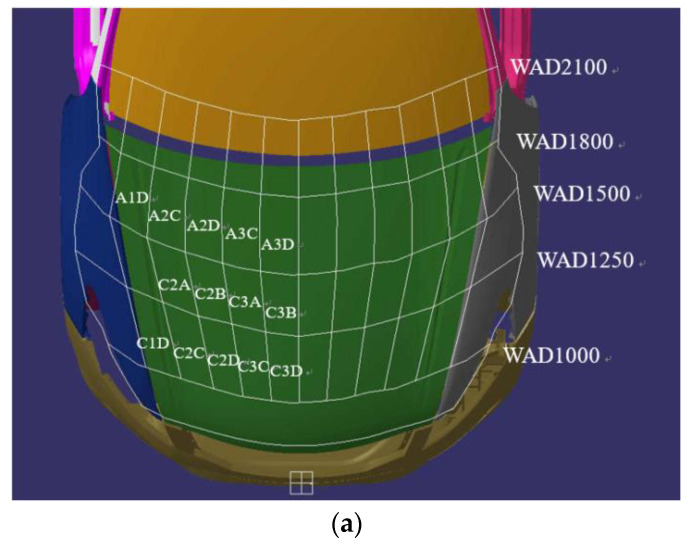
(**a**) head impact zone division, (**b**) HIC values of four different stacking sequences, and (**c**) energy absorption comparison of four different stacking sequences.

**Table 1 polymers-14-03917-t001:** CF and glass fiber reinforced hybrid composites.

Reference	Materials	Main Findings
Dong [14,15]	CF/glass fiber	Obtained the trend of the flexural strength of hybrid composites with different hybrid ratios.
Ma [16]	CF/glass fiber	Found that mixing CF into glass fiber can significantly improve the strength and modulus of the material.
Novak [17]	CF/glass fiber	Due to the high breaking elongation of glass fiber, the stability of matrix crack propagation can be improved, so that the ability of the hybrid laminate to resist impact is increased by 4 to 5 times.
Hosur [18]	CF/glass fiber	The stiffness of the hybrid ply will decrease slightly after mixing glass fiber.
Hung [19]	CF/glass fiber	Observed their failure modes.

**Table 2 polymers-14-03917-t002:** Properties of fibers.

Properties of Fibers	Reinforcement Material		Properties of Matrix	Matrix Material
	BF	CF		Epoxy/Hardener
Areal Density (g·m^−2^)	320	280	Density(g·cm^−3^)	1.12 ± 0.01/1.03 ± 0.01
Tensile Strength (MPa)	2100	4000	Compressive Strength (MPa)	124~127
Tensile Modulus (MPa)	105	240	Flexural Strength (MPa)	59~92
Density(g·cm^−3^)	2.8	1.7	Tensile Strength/MPa	59~92

**Table 3 polymers-14-03917-t003:** Comparison of different hybrid engine hood.

Evaluation Index (mm, MPa)	Steel	CFRP	B1	B2	B3
Max. Displacement (Condition 1)	1.331	1.045	1.061	1.076	1.088
Max. Displacement (Condition 2.1)	1.085	0.946	0.965	0.976	0.998
Max. Displacement (Condition 2.2)	2.313	2.263	2.296	2.327	2.351
Max. Stress (Condition 3) Max. Displacement (Condition 3)	264.8 11.80	181.3 10.79	185.5 10.97	187.3 11.13	189.7 11.36
Max. Displacement (Condition 4)	6.374	6.369	6.427	6.485	6.528
Max. Displacement (Condition 5)	1.848	1.036	1.080	1.124	1.164
First order modal (Condition 6)	25.10	27.12	26.70	26.32	25.94
Mass (kg)	16.32	9.45	9.72	9.99	10.27

**Table 4 polymers-14-03917-t004:** Comparison of results of different working conditions.

Condition	Test	FEM	Error
Installation Deformation	1.13 mm	1.045 mm	7.5%
Edge Deformation	0.98 mm	0.946 mm	3.5%
Pull-down Deformation	11.62 mm	10.79 mm	7.1%
Torsion Deformation	6.62 mm	6.369 mm	3.8%
Modal Analysis	26.2 Hz	27.1 Hz	3.4%

## Data Availability

The raw/processed data required to reproduce these findings are available upon request from the authors.

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
