# Peer review of "Carbon/Basalt Fibers Hybrid Composites: Hybrid Design and the Application in Automobile Engine Hood"

_polymers, 2022, doi:10.3390/polym14183917_

Round 1

Author Response

Please see in the attachment

Reviewer 2 Report

The manuscript is on Carbon/Basalt fibers hybrid composites for automotive hood engine applications. The manuscript is well organized, and results are interesting. Below are some comments needs to be addressed.

In lots of equations/ criterion/ model, references are missing. For example,

(a)    In page 3, provide reference for Chang-Chang fiber strength theory.

(b)     In page 4, provide reference for “ultimate strength” in different stress direction.

(c)     In page 4, provide reference for “Benzeggagh and Kenane model”.

Author Response

Please see in the attachment

Reviewer 3 Report

This paper report some innovative studies on Carbon/Basalt Fibers Hybrid Composites especially for the case of Hybrid Design and the Application in Automobile Engine Hood. The study is original and has some good literature reviews and findings. This reviewer has the following minor revision requests.

1) Literature review presented in Introduction reports several messy infos. Can it be presented in a table showing the materials, main findings, and a classification?

2) Figure 6 is very lengthy. That should be shortened.

3) Figure 7 is  a table. It is not a figure. Its colors are extremely sharp. Lighter colors are suggested.

4) The meaning of the pieces in Figure 8 is unclear. Make Mode in the first box is unclear.

5) How do you calculate the errors in Table 3?

6) The research details the performed optimization studies heavily. Conclusion does not mention the optimization studies and its benchmarking findings with errors.

Author Response

Please see in the attachment

Round 2

Reviewer 1 Report

The authors have made substuntial changes to the manuscript following review to improve it. It can now be accepted in its current form.